# Model Bridge Span Traversed by a Heavy Mass: Analysis and Experimental Verification

Georgios I. Dadoulis  and George D. Manolis *

Laboratory for Experimental Mechanics, School of Civil Engineering, Aristotle University of Thessaloniki, GR-54124 Thessaloniki, Greece; dadoulis@civil.auth.gr
* Correspondence: gdm@civil.auth.gr; Tel.: +30-2310-995663

**Abstract:** In this work, we investigate the transient response of a model bridge traversed by a heavy mass moving with constant velocity. Two response regimes are identified, namely forced vibrations followed by free vibrations as the moving mass goes past the far support of the simply supported span of the bridge. Despite this being a classical problem in structural dynamics, there is an implicit assumption in the literature that moving loads possess masses that are at least an order of magnitude smaller than the mass of the bridge span that they traverse. This alludes to interaction problems involving secondary systems, whose presence does not alter the basic characteristics of the primary system. In our case, the dynamic properties of the bridge span during the passage of a heavy mass change continuously over time, leading to an eigenvalue problem that is time dependent. During the free vibration regime, however, the bridge recovers the expected dynamic properties corresponding to its original configuration. Therefore, the aim here is the development of a mathematical model whose numerical solution is validated by comparison with experimental results recovered from an experiment involving a scaled bridge span traversed by a rolling mass. Following that, the target is to identify regions in the transient response of the bridge span that can be used for recovering the bridge's dynamic properties and subsequently trace the development of structural damage. In closing, the present work has ramifications in the development of structural health monitoring systems applicable to critical civil engineering infrastructure, such as railway and highway bridges.

**Keywords:** bridge model; moving loads; modal analysis; experimental verification; transient response; frequency spectra



## 1. Introduction

The subject of moving loads on beams dates back to the 19th century, following the development of railways that necessitated the construction of metallic bridges across rivers, valleys and other types of irregular topography. At the same time, there was a parallel effort in developing mathematical models for this engineering type of problem, see, for instance, the early treatise by Renaudot [1] on representing a mass rolling over a beam. Starting from the 1960s onwards with the seminal work by Fryba [2], much work, both analytic and numerical, has been carried out on variations of the basic problem of a single load moving with constant speed over a single span. These include the cases of multiple spans, multiple moving loads, loads that accelerate or decelerate etc. Furthermore, the point load itself has evolved to become a structural sub-system by itself in possession of a mass, a stiffness and a damper in order to better model the passing vehicle's suspension system, see, for instance, Liu et al. [3]. Further studies along these lines were performed by Green and Cebon [4], who examined a simply supported highway bridge traversed by a single degree-of-freedom vehicle model and derived six non-dimensional parameters that quantified the degree of interaction between vehicle and bridge. The same authors expanded their original study to include the dynamic bridge response to a given set of wheel loads and also performed field measurements on a highway bridge to validate their

numerical model [5]. In his doctoral dissertation, Johansson [6] gives a good summary on the hierarchy of the mathematical models used for train–track–bridge interaction and goes on to discuss probabilistic dynamic analysis of single bridges, as well as bridge networks, for high-speed railway traffic. An excellent exposition on the vibrations of structures under moving inertial loads with applications to railway problems is given in the monograph by Bajer and Dyniewicz [7], who go on to develop space–time finite elements for the vibrations of both Bernoulli–Euler and Timoshenko beams. Finally, much information on the dynamics of structures and structural components can be found in the treatises by Rao [8] and Kausel [9] from a mechanical engineering and a structural engineering point of view, respectively.

Over time, the monitoring of existing railway and highway bridges for the purpose of retrofit and rehabilitation has become of paramount importance and falls within the realm of critical infrastructure maintenance [10]. In reference to our work, which can be viewed as a background for developing structural health monitoring systems for bridges, we need to distinguish between various damage sources that lead to possible failure. The most important one is probably structural damage, encompassing the loosening of support integrity, cracking and disintegration of the main load-bearing system, the lack of fit resulting from temperature variations etc. However, there are further sources of damage in bridges due to dynamic and other categories of external loads. Among much of the work that appears in the literature, we mention here a numerical study on the response and subsequent failure modes of R/C bridge columns under vehicle collisions, which are classified as flexural failure, shear failure and punching shear damage [11]. Similarly, the detection of early-stage corrosion in slender steel members used for supporting structures, such as bridges, by employing photoacoustic fiber-optic sensors involving a transmitter and a receiver configuration was reported in [12]. More specifically, the finite element method was applied for the numerical simulation of the propagation of ultrasonic waves on steel rod models. Recently, a damage detection method for railway bridges was proposed by Azim and Gul [13], who used principal component analysis of the bridge dynamic strain response. Thus, the time history response of truss-type bridges was compared between the baseline and damaged conditions using finite element simulations with artificial noise inserted into the models, and the deviation in the strain response was correlated to the change in stiffness brought about by damage.

### 1.1. Structural Health Monitoring Issues

Structural health monitoring (SHM) plays an important role in assuring the continuous operation and functionality of modern infrastructure. In general, infrastructure can be classified as the built environment plus the necessary networks for energy (power grids, wind turbines and pipelines), for water supply and disposal, for communications (antennas, transmitters and cables) and for transportation (roads, bridges, tunnels and railways). Essentially, networks form a special category of engineering construction whose operation must remain unhindered over time inasmuch as possible. Furthermore, concerns regarding infrastructure resilience have led to the classification of 'critical infrastructure' for networks that must be continuously maintained to help with the rapid response and recovery capabilities of a given geographical area in the event of natural disasters [14]. Traditionally, the structural maintenance of critical infrastructure networks has been conducted using non-destructive testing (NDT) evaluation practices, such as on-site visual inspections and measurements. However, these rely on the availability and judgment of skilled personnel, whose absence results in discontinuities in the inspection protocol, as well as in bias regarding the structural condition of the network in question. Advances in sensor technologies over the past decades [15] have enabled the development of the SHM paradigm, whereby critical infrastructure networks can be continuously monitored over time in terms of their overall performance and response to the working loads plus other environmentally induced loads [16]. Thus, data streams generated by this continuous mon-

itoring can be processed with the aid of artificial intelligence (AI) algorithms that will allow the managing authorities to reach rational conclusions regarding network operations [17].

The collection of data in SHM systems is performed in tandem with data processing to extract useful information on the structural condition of the network in question. These data processing methods can be combined with numerical modeling in either of two ways [15]: (a) educated assumptions on material and structural properties of the monitored structure, known as 'model-based' monitoring, or (b) relationships between structural response data sets regardless of the physics underlying the structural response, known as 'model-free' monitoring. Although the majority of SHM strategies are associated with the latter category, 'model-based' data processing yields insights into the relationship between the structural response and structural behavior. To this end, the focus of our work is to develop analytical tools that are calibrated against experimental evidence and can rapidly and accurately compute the anticipated structural response in order to compare it against the measured one. We focus here on single-span bridges under moving loads whose mass is comparable to that of the supporting deck, resulting in an alteration of the dynamic properties of the bridge during the passage of these loads.

*1.2. External Load Categories*

Vibration tests on civil engineering structures, such as bridges, have been running for many years now, and it has become clear that environmental parameters affect their dynamic behavior. For instance, the elasticity modulus decreases with increasing temperature and with long-term exposure to humidity. Since damage detection is one of the main aims of vibration monitoring, the basic premise that holds is that loss of stiffness observed in time shifts the eigenfrequencies to the low end of the spectrum. However, changes due to damage can be completely masked by changes due to predictable environmental conditions. The main problem when analyzing vibration measurements as a tool for SHM is thus separating abnormal changes from normal changes in the dynamic response of the structure in question [10]. The former changes are caused by conditions such as temperature, humidity, wind and ground motions, while the latter ones are due to an overall degradation of stiffness. It is clear that the normal changes should not trigger a false alarm in the monitoring system, whereas the abnormal changes must be detected because they may be critical for safety purposes.

As an example, Peeters at al. [18] discuss the Z24 bridge in Switzerland that was monitored for one year before it was artificially damaged. Black box models were then built from the undamaged bridge data by applying system identification techniques. These models described the variations in the natural frequencies as a function of temperature and were used for gauging new incoming data. If a natural frequency exceeded certain confidence intervals established by the model, then it was probable that another cause, perhaps damage, drove this change. More specifically, an automatic modal analysis procedure based on stochastic subspace identification was proposed to extract the modal parameters from stabilization diagrams without any user interaction. By carefully inspecting the available data, the physical phenomenon behind a typical bilinear relation between frequency and temperature was recovered. Due to the relatively large amount of recorded data, a more detailed data analysis was possible as compared to the classical statistical regression analysis. A unique data set could then be used to validate the proposed method, in the sense that measurements were available year round.

As another application example, Sohn et al. [19] introduced a linear adaptive filter that discriminated changes in modal parameters due to temperature from those caused by structural damage by using data from the Alamosa Canyon Bridge in the state of New Mexico. This filter solves the eigenvalue problem using conventional methods but is able to adapt its prediction to the natural frequencies of the structure using a pre-determined time–temperature profile. Thus, it becomes possible to discriminate changes in modal parameters due to temperature from those caused by other environmental factors or by structural damage. Specifically, when the measured frequencies move outside the predicted

confidence intervals, the system can provide a reliable indication that structural changes are likely caused by factors other than heat. Changes in the natural frequencies are found to correlate linearly with temperature readings from different locations on the bridge. The filter used spatial and temporal temperature distributions to determine changes in the first and second natural frequencies. Furthermore, it was possible to account for non-stationarity in the natural frequencies caused by environmental factors by using a linear filter with two spatially separated and two temporally separated temperature measurements that reproduced the variation of the natural frequencies from a first data set.

## 2. Experimental Setup

We first discuss the experimental setup shown in Figure 1 for a bridge model traversed by a rolling mass as a way of introducing the mathematical formulation that follows. Specifically, the model girder was a steel section HEA100 [20] of length $L = 6.13$ m, cross-sectional area $A = 21.24 \times 10^{-4}$ m$^2$, moment of inertia about the strong axis of $I = 349.2 \times 10^{-8}$ m$^4$, modulus of elasticity $E = 200 \times 10^6$ kPa and mass density $\rho = 7.85$ tn/m$^3$. The travelling point mass had a magnitude $m = 34.6$ kg, which could be increased to higher values, as compared to a girder mass $M = 102.2$ kg. The point mass moved with a constant, reference speed of $v = 1.0$ m/s, meaning that the time to traverse the span was $t = 6.13$ s, past which the girder was under a state of free vibrations. Again, this speed was adjustable, meaning that it could be increased by faster rotation of the winch that pulled the mass across the girder's span using a wire. The girder's ends were placed on short steel columns with flexible pads as an interface; thus, the case of a simply supported beam could be recovered when the pads were removed. When support rigidity was relaxed, it was estimated that the pads could be represented by springs of modulus $K_1$, $K_2$ at the right and left supports, respectively. Values for these springs were estimated to range from $K_1 = K_2 = 1000$ kN/m to $K_1 = K_2 = 100$ kN/m. The damping parameter for the bridge model was estimated as $\zeta = 0.0001$ for all modes of vibration. Furthermore, the radius of gyration of the cross-section is $r = \sqrt{I/A} = 0.041$ m, which implies a slender beam without shearing effects; thus, it is unnecessary to resort to the Timoshenko beam theory.

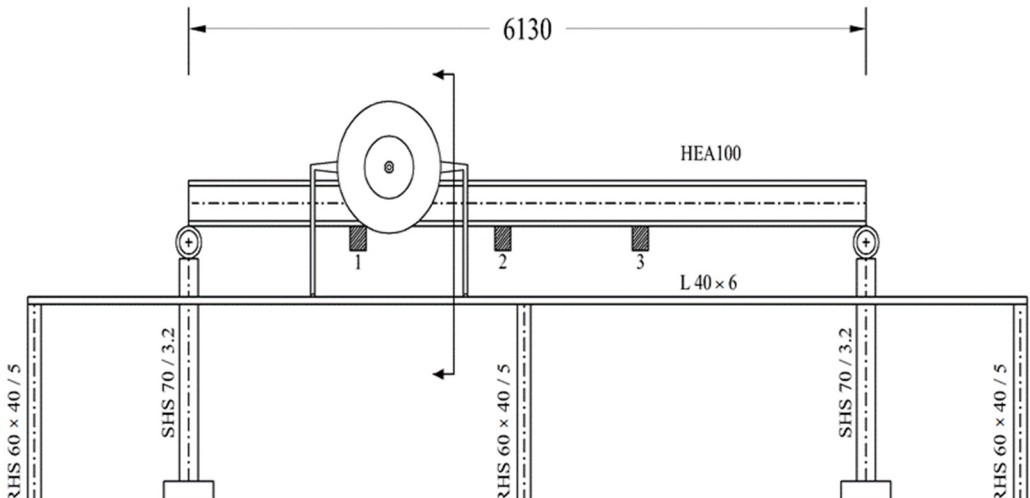

**Figure 1.** Experimental setup for a simply supported girder of length $L = 6130$ mm under a heavy mass moving with constant speed and accelerometers placed at $L/4$, $7L/16$ and $2L/3$. Note: The undercarriage acts as a railguide for the moving mass.

## 3. Beam Models for the Bridge Deck on Flexible Supports

We now examine both the conventional Bernoulli–Euler beam representation and the higher order Rayleigh beam theory to check the former's validity for the expected frequency range relevant to an experimental setup with a heavy mass moving along a model bridge span at constant speed. The equation of motion for a Rayleigh beam underdoing free vibrations [9] is

$$r^2 \frac{\partial^4 w(x,t)}{\partial x^2 \partial t^2} = a^2 \frac{\partial^4 w(x,t)}{\partial x^4} + \frac{\partial^2 w(x,t)}{\partial t^2} \tag{1}$$

where $w(x,t)$ is the transverse displacement and coefficient $a = \sqrt{EI/\rho A}$. The values for all problem parameters can be found in Section 2. In order for the mixed inertia term at the *RHS* of the above equation to become sizeable, the speed by which the mass moves across the span must be high so that high-frequency vibrations are generated; otherwise, it is negligible. For boundary conditions that correspond to simple supports, the characteristic equation that yields the eigenfrequencies of vibration is

$$\sin(m_1 L)\sinh(m_2 L) = 0$$

$$m_1 = \sqrt{\frac{\omega^2 r^2}{2a^2} + \sqrt{\frac{\omega^4 r^4}{4a^4} + \frac{\omega^2}{\alpha^2}}} \qquad m_2 = \sqrt{-\frac{\omega^2 r^2}{2a^2} + \sqrt{\frac{\omega^4 r^4}{4a^4} + \frac{\omega^2}{\alpha^2}}}$$

For the simpler Euler–Bernoulli beam, the equation of motion is now

$$EI \frac{\partial^4 w(x,t)}{\partial x^4} + \rho A \frac{\partial^2 w(x,t)}{\partial t^2} = 0 \tag{2}$$

The characteristic equation for a simply supported case is $\sin(\kappa L) = 0$, $\kappa = n\pi/L$, $n = 1, 2, \ldots$. In Table 1 which follows, we compare the first seven eigenfrequencies as computed for the Bernoulli–Euler and Rayleigh beams. Given that the frequency spectra for the model beam under the heavy moving load do not show appreciable information past 100 Hz, we adopt the simpler Bernoulli–Euler beam model.

**Table 1.** Euler–Bernoulli and Rayleigh beam eigenfrequencies (*Hz*) for simply supported beam.

| Bernoulli–Euler Beam | | | | | | |
|---|---|---|---|---|---|---|
| 8.56 | 34.23 | 77.01 | 136.9 | 213.9 | 308.0 | 419.3 |
| **Rayleigh Beam** | | | | | | |
| 8.55 | 34.20 | 76.86 | 136.4 | 212.8 | 305.7 | 414.9 |

Next, we look at the eigenfunctions of the simply supported Bernoulli–Euler beam, which are given as $\Phi_n(x) = \sqrt{2/M} \, \sin\left(\frac{n\pi x}{L}\right)$, $n = 1, 2, \ldots$, where $M = \rho A L$ is the total mass of the beam. In the case of flexible end supports, the computational procedure is described in the Appendix A, and the relevant figures are Figure 2b,c.

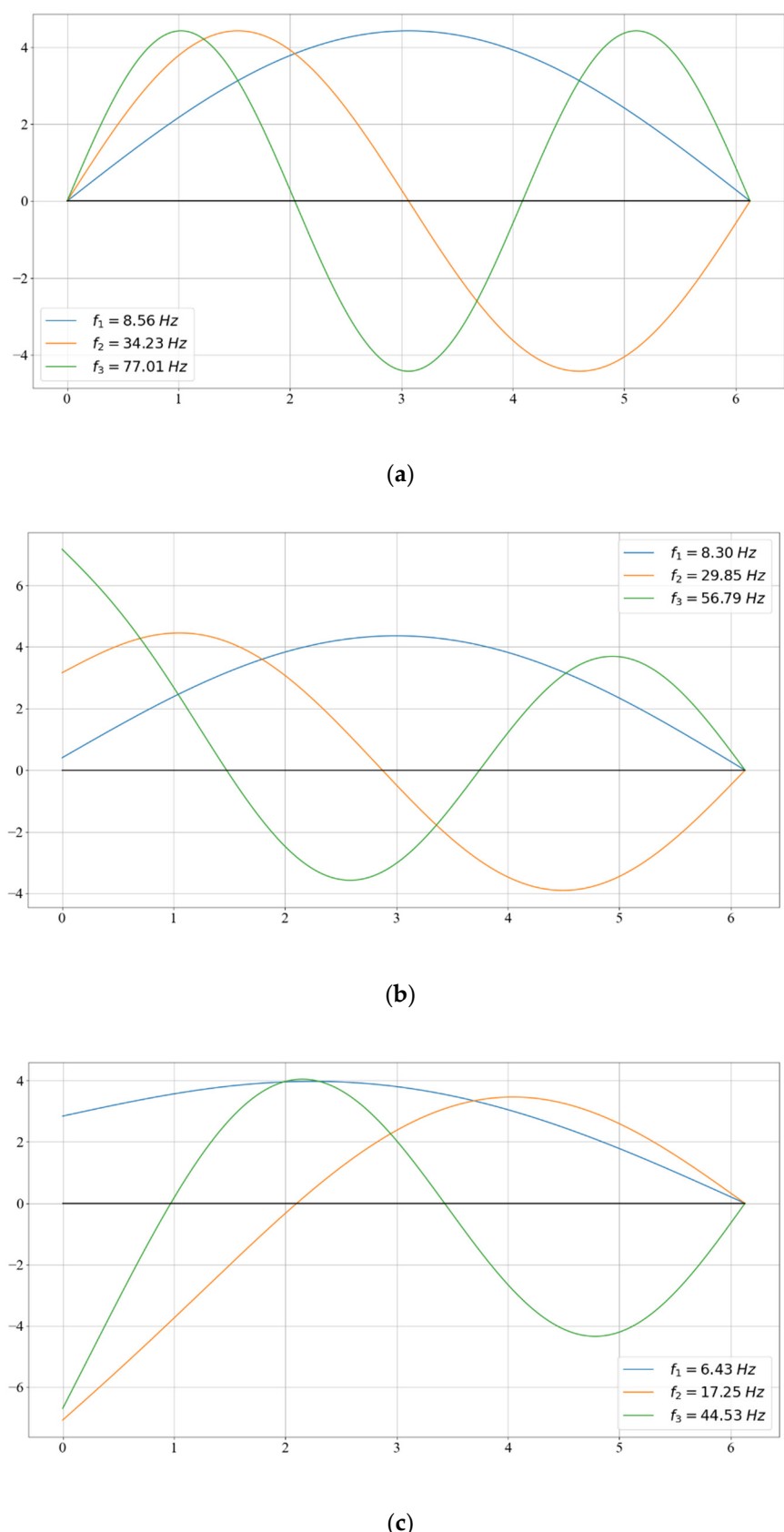

**Figure 2.** The first three eigenfrequencies and eigenfunctions of the bridge deck for (**a**) simple supports at the two ends; (**b**) flexible left support with $K_1$ = 1000 kN/m and $K_2$ = ∞; (**c**) flexible left support with $K_1$ = 100 kN/m and $K_2$ = ∞.

## 4. Mathematical Models for Moving Loads

### 4.1. Stationary Mass Reference Case

Assume that a mass $m(tn)$ is placed at station $x_0$ along the span of the beam. If the mass is large as compared to the total mass $M$ of the beam, i.e., we no longer have a secondary system placed on a primary one, it is expected that the modal characteristics of the combined system will change. The equation of motion now reads as follows:

$$\rho A\,\ddot{w} + EIw'''' = -m\ddot{w}\,\delta(x - x_0) \tag{3}$$

where $EI$ and $\rho A$, respectively, are the girder's flexural stiffness and its mass per unit length, while dots and primes, respectively, indicate differentiation with respect to time $t$ and coordinate $x$. Finally, $\delta$ is the Dirac delta function. If the boundary conditions are homogeneous, it is possible to use separation of variables in the form of the generalized coordinates $q_i(t)$ and express the transverse displacement as

$$w(x,t) = \sum_{n=0}^{\infty} \Phi_n(x)q_n(t)$$

where the summation convention is implied for repeated indices. Inserting this expression in Equation (3) and integrating along the beam's length, we obtain the following $3 \times 3$ matrix for $n = 3$ modes, where $\omega_n$ $(Hz)$ are the eigenfrequencies of the beam deck alone:

$$\left( \begin{bmatrix} 1 & 0 & 0 \\ 0 & 1 & 0 \\ 0 & 0 & 1 \end{bmatrix} + m \begin{bmatrix} \Phi_1(x_0)\Phi_1(x_0) & \Phi_1(x_0)\Phi_1(x_0) & \Phi_1(x_0)\Phi_1(x_0) \\ \Phi_2(x_0)\Phi_1(x_0) & \Phi_2(x_0)\Phi_1(x_0) & \Phi_2(x_0)\Phi_1(x_0) \\ \Phi_3(x_0)\Phi_1(x_0) & \Phi_3(x_0)\Phi_1(x_0) & \Phi_3(x_0)\Phi_1(x_0) \end{bmatrix} \right) \begin{bmatrix} \ddot{q}_1 \\ \ddot{q}_2 \\ \ddot{q}_3 \end{bmatrix} + \begin{bmatrix} \omega_1^2 & 0 & 0 \\ 0 & \omega_2^2 & 0 \\ 0 & 0 & \omega_3^2 \end{bmatrix} \begin{bmatrix} q_1 \\ q_2 \\ q_3 \end{bmatrix} = \begin{bmatrix} 0 \\ 0 \\ 0 \end{bmatrix} \tag{4}$$

Returning to the generalized coordinates in the presence of the stationary mass, we assume that harmonic vibrations still hold so that $\ddot{q} = \overline{\omega}_n{}^2 q_n$, where $\overline{\omega}_n$ are now the eigenfrequencies of the combined system of the beam plus the mass. Carrying out computations for the model bridge defined in the previous section, we present results for three cases in Table 2 below. We observe that the presence of a fixed mass results in a more flexible system leading to a drop in the eigenfrequencies. It should be noted, however, that when the mass passes a station for which an eigenfunction has a zero crossing, the corresponding eigenfrequency is unchanged.

**Table 2.** Eigenfrequencies (Hz) of the model bridge comprising a primary system (beam) and a secondary system (fixed mass).

| Eigenfrequency Number | Reference Beam without a Stationary Mass | Beam with a Stationary Mass $m$ = 34.6 kg at Station $x_0$ = L/4 | Beam with a Stationary Mass $m$ = 34.6 kg at Station $x_0$ = L/3 | Beam with a Stationary Mass $m$ = 34.6 kg at Station $x_0$ = L/2 |
|---|---|---|---|---|
| 1 | 8.56 | 7.39 | 6.97 | 6.61 |
| 2 | 34.22 | 26.43 | 27.87 | 34.22 |
| 3 | 77.01 | 66.55 | 77.01 | 59.46 |

### 4.2. Moving Load Case

When the moving mass is small compared to the mass of the beam that it traverses with velocity $v$ (m/s), it can be considered as a moving load that does not change the dynamic properties of the beam. The equation of motion in the presence of damping $c$ (kN·s/m) is now

$$\rho A\,\ddot{w} + c\,\dot{w} + EIw'''' = m\ddot{w}\,\delta(x - vt) \tag{5}$$

where $g$ is the acceleration of gravity. The analytical solution for the simply supported beam is recovered by the separation of variables plus the use of Duhamel's convolution integral for handling the load term in the *RHS* of the above equation. Specifically, we have

$$w(x,t) = \sum_{n=1}^{\infty} \frac{2mg}{M\omega_{d,n}} \sin\left(\frac{n\pi}{L}\right) \cdot \int_0^t e^{-\xi_n \,\omega_n(t-\tau)} \sin(\omega_{d,n}(t-\tau)) \sin(k_n v\tau) \, d\tau \quad (6)$$

In the above, $\omega_{d,n} = \omega_n \sqrt{1-\xi^2}$ is the damped natural frequency assuming damping is the same for all modes $n = 1, 2, \ldots$ We can now define a critical velocity for the simply supported Bernoulli–Euler beam representing the bridge girder as follows [6]:

$$v_{cr} = \frac{\pi}{L}\sqrt{\frac{EI}{\rho A}} = 104.9 \text{ m/s} \quad (7)$$

If the velocity of the moving load is less than $v_{cr}$, the maximum value attained by the transverse displacement $w_{max}$ occurs when the load is still on the bridge's span. Otherwise, $w_{max}$ occurs in the free vibration regime, i.e., after the moving load has left the span. Some numerical results are given in Figure 3 below for the model bridge used in the experiment (Figure 1). Note that the vertical red line delineates the two phases of vibration, i.e., when the load is traversing the beam's span and when it has left it. Specifically, the value of the moving load is $P_0 = mg = 34.6 \times 9.81 = 0.34$ kN, and the displacement is evaluated at the center of the span $x = L/2$ for three values of velocity, namely $v = 64.7$ m/s $< v_{cr}$, $v = v_{cr} = 104.9$ m/s and $v = 160$ m/s $> v_{cr}$. We observe that the maximum transverse displacement of 4 mm occurs for the subcritical velocity, while for the other two cases, this value drops to 3.5 mm and 2.8 mm, respectively.

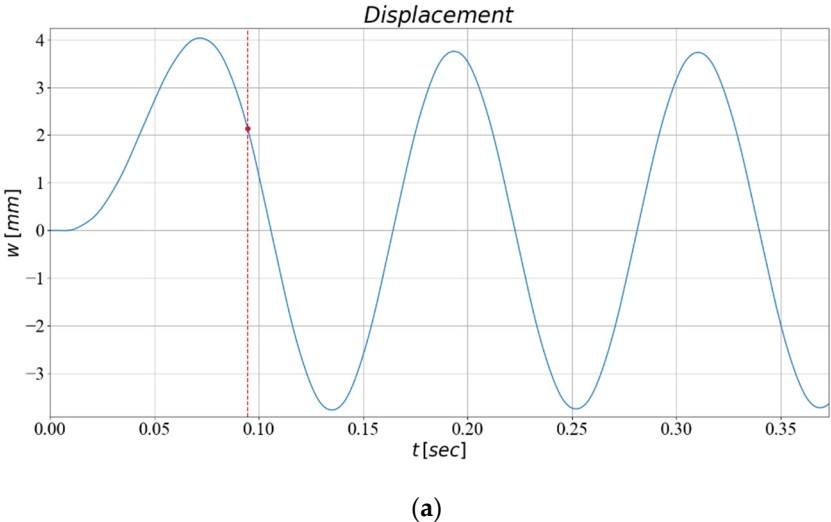

(**a**)

**Figure 3.** *Cont.*

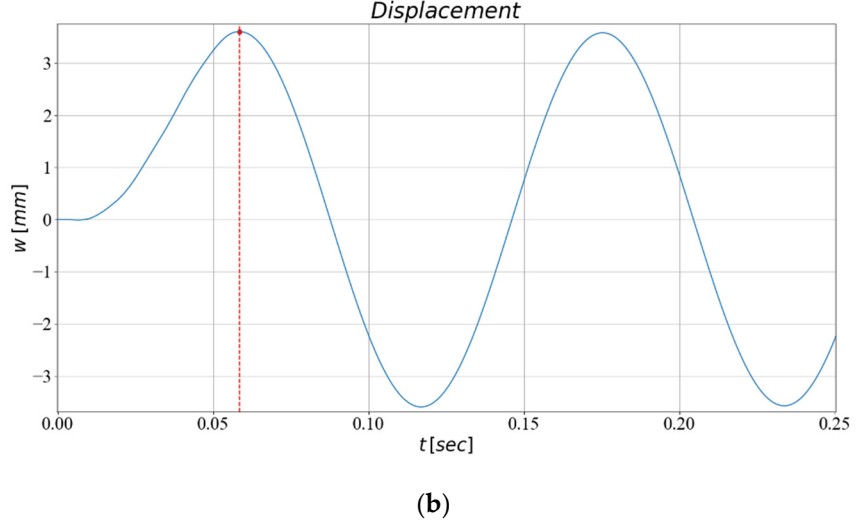

**(b)**

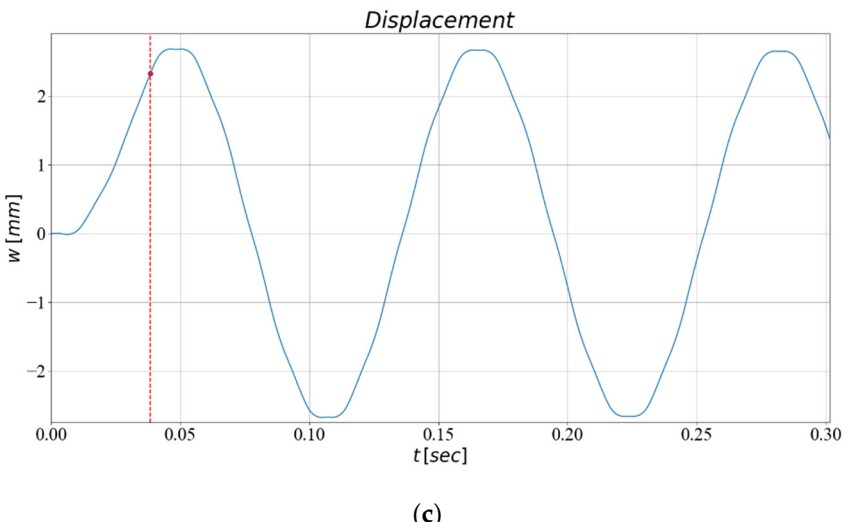

**(c)**

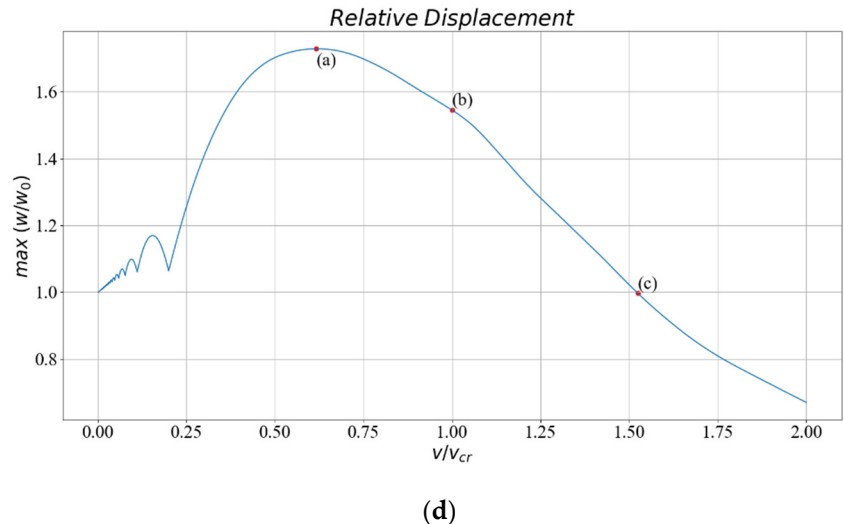

**(d)**

**Figure 3.** Transverse beam displacement at L/2 for (**a**) subcritical moving load velocity of 64.7 m/s, (**b**) critical velocity of 104.9 m/s, (**c**) supercritical velocity of 159.5 m/s and (**d**) maximum transverse displacement for all three load velocities. Note: The three red dots associate with cases (**a–c**).

### 4.3. Moving Heavy Mass Case

The governing equation of motion for a single-span bridge girder under a moving point mass, which, however, includes the mass's inertia effect, is

$$\rho A\, \ddot{w} + c\, \dot{w} + EIw'''' = m\ddot{w}\left(g - \frac{d^2 w(x,t)}{dt^2}\right)\delta(x - vt) \tag{8}$$

As shown in Figure 1, $m$ is the mass moving across the span with constant velocity $v$ and $w(x,t)$ is again the transverse displacement. In reference to the *RHS* of the above equation, when moving mass $m$ changes its position with time, the material derivative is written as

$$\frac{d^2 w}{dt^2} = \frac{\partial^2 w}{\partial t^2} + 2v\frac{\partial^2 w}{\partial x \partial t} + v^2\frac{\partial^2 w}{\partial x^2} \tag{9}$$

where the first term is the rate of change of the girder's displacement and the remaining terms refer to the rate of change of the mass vertical motion as it moves across the span.

Starting with the eigenvalue problem for the Bernoulli–Euler beam representation of the bridge girder resting on elastic springs $K_1$, $K_2$ the transverse displacement is written in terms of the generalized coordinates as $w(x,t) = \Phi_n(x)q_n(t)$. Following Renaudot [1] and taking advantage of the normality property of the eigenfunctions $\Phi_n$, we recover the equation of motion in the form of a $3 \times 3$, time-dependent matrix system by retaining these eigenmodes:

$$[M(t)]\{\ddot{q}(t)\} + [D(t)]\{\dot{q}(t)\} + [E(t)]\{q(t)\} = \{P(t)\} \tag{10}$$

where

$$[M] = \begin{bmatrix} 1 & 0 & 0 \\ 0 & 1 & 0 \\ 0 & 0 & 1 \end{bmatrix} + m\begin{bmatrix} \Phi_1(vt)\Phi_1(vt) & \Phi_1(vt)\Phi_2(vt) & \Phi_1(vt)\Phi_3(vt) \\ \Phi_2(vt)\Phi_1(vt) & \Phi_2(vt)\Phi_2(vt) & \Phi_2(vt)\Phi_3(vt) \\ \Phi_3(vt)\Phi_1(vt) & \Phi_3(vt)\Phi_2(vt) & \Phi_3(vt)\Phi_3(vt) \end{bmatrix}$$

$$[D] = 2\xi\begin{bmatrix} \omega_1 & 0 & 0 \\ 0 & \omega_2 & 0 \\ 0 & 0 & \omega_3 \end{bmatrix} + 2mv\begin{bmatrix} \Phi_1(vt)\Phi_1'(vt) & \Phi_1(vt)\Phi_2'(vt) & \Phi_1(vt)\Phi_3'(vt) \\ \Phi_2(vt)\Phi_1'(vt) & \Phi_2(vt)\Phi_2'(vt) & \Phi_2(vt)\Phi_3'(vt) \\ \Phi_3(vt)\Phi_1'(vt) & \Phi_3(vt)\Phi_2'(vt) & \Phi_3(vt)\Phi_3'(vt) \end{bmatrix}$$

$$[E] = \begin{bmatrix} \omega_1^2 & 0 & 0 \\ 0 & \omega_2^2 & 0 \\ 0 & 0 & \omega_3^2 \end{bmatrix} + mv^2\begin{bmatrix} \Phi_1(vt)\Phi_1''(vt) & \Phi_1(vt)\Phi_2''(vt) & \Phi_1(vt)\Phi_3''(vt) \\ \Phi_2(vt)\Phi_1''(vt) & \Phi_2(vt)\Phi_2''(vt) & \Phi_2(vt)\Phi_3''(vt) \\ \Phi_3(vt)\Phi_1''(vt) & \Phi_3(vt)\Phi_2''(vt) & \Phi_3(vt)\Phi_3''(vt) \end{bmatrix}$$

$$\{P\} = mg\begin{Bmatrix} \Phi_1(vt) \\ \Phi_2(vt) \\ \Phi_3(vt) \end{Bmatrix}$$

As before, $\omega_n(Hz)$ are the eigenfrequencies of the bridge deck alone.

## 5. Numerical Implementation for the Heavy Mass Case

The solution of the second-order matrix differential equation system of Equation (10) can be accomplished using the Runge–Kutta method of order four *(RK4)*, following conversion to a first-order matrix differential equation system of order $2n = 6$,

$$\frac{d}{dt}\begin{Bmatrix} y_n(t) \\ z_n(t) \end{Bmatrix} + \begin{bmatrix} A_{11} & A_{12} \\ A_{21} & A_{22} \end{bmatrix}\begin{Bmatrix} y_n(t) \\ z_n(t) \end{Bmatrix} = \begin{Bmatrix} P_{1n}(t) \\ P_{2n}(t) \end{Bmatrix}, \quad \left\{y_n(t) = q_n(t),\ z_n(t) = \frac{d}{dt}q_n(t)\right\} \tag{11}$$

The solution scheme was subsequently upgraded to an adaptive Runge–Kutta method of order five *(RK5)* for better accuracy. Specifically, for a first-order differential equation, we use *RK4* and compute error $e(h_1)$ for step $h_1$ and repeat the procedure to compute error $e(h_2)$ for step $h_2$, and then the relation between the steps and errors can be estimated. Thus, by prescribing a tolerance $\varepsilon$ and running the *RK5* for an initial step $h_1$, the new step for re-computing is $h_2 = 0.90\, h_1\left(\frac{\varepsilon}{e(h_1)}\right)^{\frac{1}{5}}$, and this process is repeated every subsequent step.

### 5.1. Numerical Details

Equation (10) may look like the standard equation of motion for a multiple-degree-of-freedom (DOF) system, but there are some important differences. Firstly, all system matrices are time dependent and, with the exception of the mass matrix, are non-symmetric. This may not be a problem for solvers that can invert non-symmetric matrices, but it is always possible to decompose them into symmetric and-antisymmetric parts, i.e.,

$$[D] = [C] + [G], \ [E] = [K] + [S] \tag{12}$$

In the above decomposition, $[C]$ is the symmetric damping matrix of the combined bridge moving mass system, while $[G]$ is the antisymmetric gyroscopic matrix. Likewise, $[K]$ is the standard stiffness matrix of the combined system, while $[S]$ is antisymmetric and referred to as the circulation matrix.

The solution of Equation (10) can only be accomplished numerically using time stepping. However, the appropriate time step $h$ is difficult to estimate but essential in minimizing computational time. For instance, high crossing velocities of the mass result in high-frequency vibrations that require a small $h$ and vice versa for low crossing velocities. It was therefore necessary to upgrade the *RK4* to an adaptive *RK5* for better accuracy, see Dormand and Prince [21]. In essence, *RK4* is used as an intermediate calculation before switching to *RK5*, and this is carried out continuously throughout the time stepping procedure. The error at every time step is actually a vector of errors committed for each variable. For instance, if we have two DOFs $y_1$ and $y_2$, then the error vector is

$$E_n(h) = \begin{bmatrix} E_1(h) \\ E_2(h) \end{bmatrix} = \begin{bmatrix} y_{1, \ RK5}(t+h) - y_{1, \ RK4}(t+h) \\ y_{2, \ RK5}(t+h) - y_{2, \ RK4}(t+h) \end{bmatrix}, \ n = 1, 2 \tag{13}$$

Two criteria are available for error $e(h)$ estimation, which, for the above case, are

$$e(h) = max|E_n(h)|, \ e(h) = \sqrt{\frac{1}{n} \sum_{i=1}^n E_i(h)} \tag{14}$$

The error estimate $e(h)$ is then used to adjust the time step using a present tolerance $\varepsilon$. Starting with the ratio $\frac{e(h_1)}{e(h_2)} \approx \left(\frac{h_1}{h_2}\right) \approx \left(\frac{h_1}{h_2}\right)^5$ and setting the tolerance based on $e(h_2)$, we recover the desired time step value as $h_2 \approx h_1 \left(\frac{\varepsilon}{e(h_1)}\right)^{\frac{1}{5}}$.

### 5.2. Numerical Results

We note that in order to compute acceleration time histories from the velocities, a simple finite-difference scheme is used as $\ddot{w}(t) = \frac{\{\dot{w}(t+h) - \dot{w}(t)\}}{h}$, where $h$ is the smallest time step value calculated from the *RK5* iterations. Figure 4 depicts the forced vibration response of the bridge girder at recording station $x = L/2$ for the support cases mentioned. We note the pivotal role played by the support conditions on the girder's response, with the left-hand side spring being soft while that at the right-hand side being stiff.

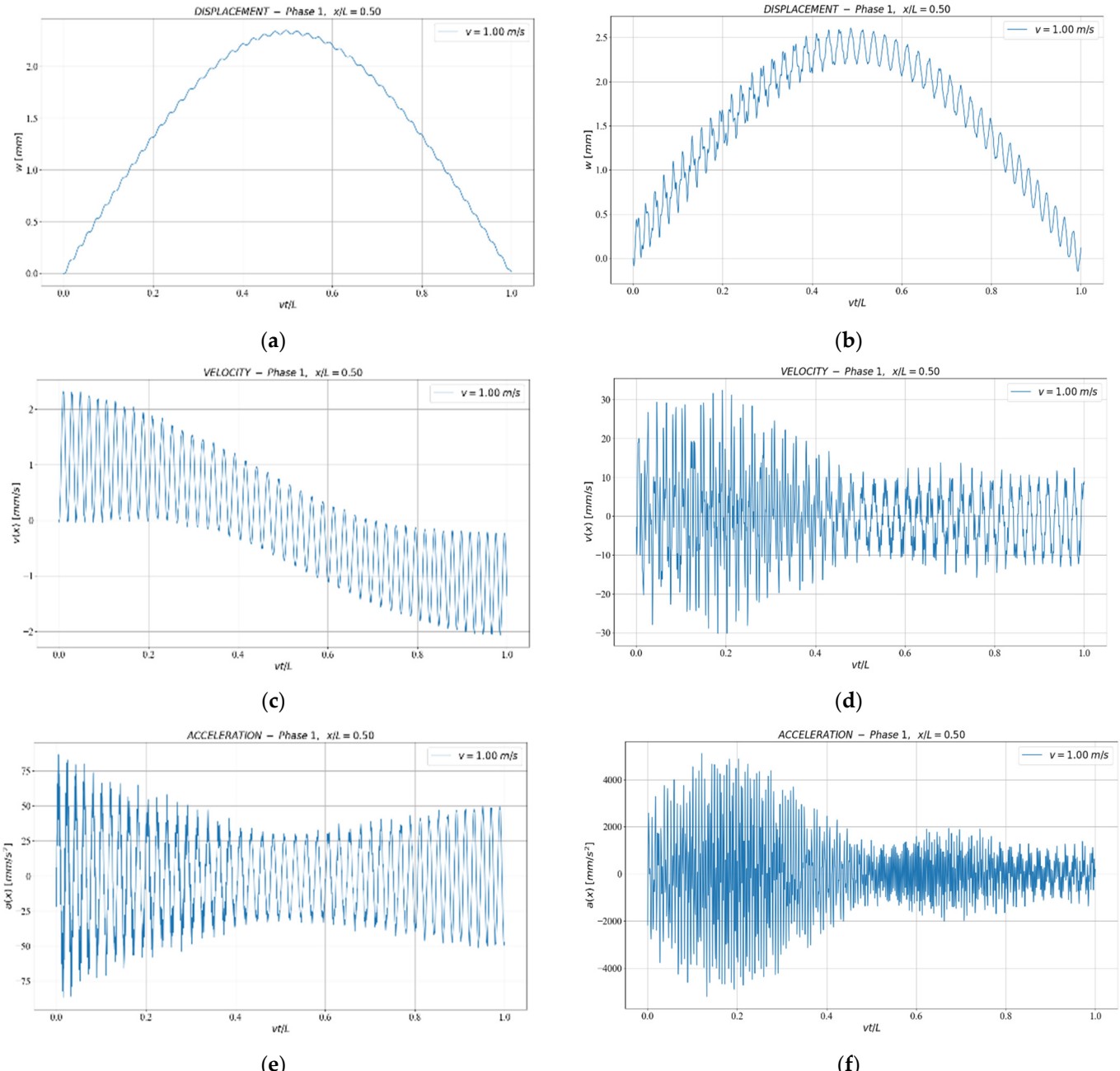

**Figure 4.** Moving point mass across the model steel girder with simple supports: left column (**a,c,e**) plots the displacement, velocity and acceleration at station $x = L/2$ for rigid supports; right column (**b,d,f**) plots the displacement, velocity and acceleration at station $x = L/2$ for elastic supports.

## 6. The Effect of a Moving Mass Inertia

In this section, we examine both the time histories and Fourier spectra for the bridge deck displacement near the center at the station $x = \frac{7L}{16}$, as recovered from a numerical solution of the analytical results developed in Section 4. More specifically, mass $m$ is analyzed both as a moving load (Section 4.2) and as a moving mass (Section 4.3) representation. Figure 5 clearly shows that for a sizeable mass relative to the bridge deck as defined by mass ratio $R = \frac{m}{M} = \frac{38.6}{102.2} = 0.38$, and at relatively low speeds, there are few differences between the numerical results derived from these two representations in the time domain. On the contrary, Figure 6 juxtaposes the difference observed when the mass becomes very heavy $\left(R = \frac{500}{102.2} = 4.89\right)$ and moves at high speed.

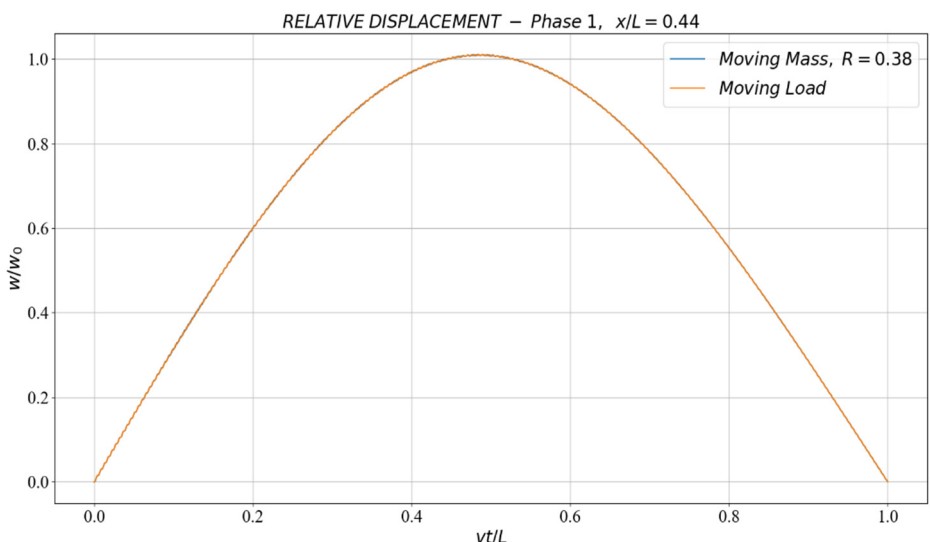

**Figure 5.** Forced vibration time history of the deck displacement at station $x = 7L/16$ of the bridge for a light $m = 38.60$ kg mass traversing at a low speed of $v = 0.247$ m/s.

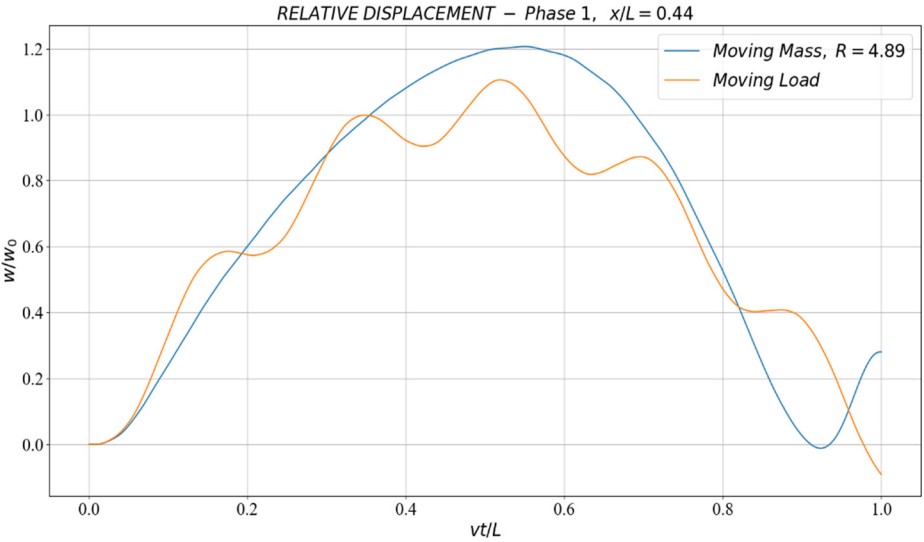

**Figure 6.** Forced vibration time history of the deck displacement at station $x = 7L/16$ of the bridge for a heavy $m = 500$ kg mass traversing at a high speed of $v = 10.0$ m/s.

Following up with an FFT of the above displacement time histories, we observe in Figure 7 the reason for divergence between the moving load and moving mass cases, even for the case in Figure 5: the latter representation yields a system whose dynamic properties change as the mass moves across, i.e., for the forced vibration part of the analysis. The first eigenfrequency of the primary system, i.e., the bridge deck, has a well discernible value of $f_1 = 8.56$ Hz, while if the analysis takes into account the combined system, this frequency is now diffused and registers values in the 6.54 Hz $< f_1 <$ 8.52 Hz range. This alludes to a more flexible system without well-defined eigenfrequencies.

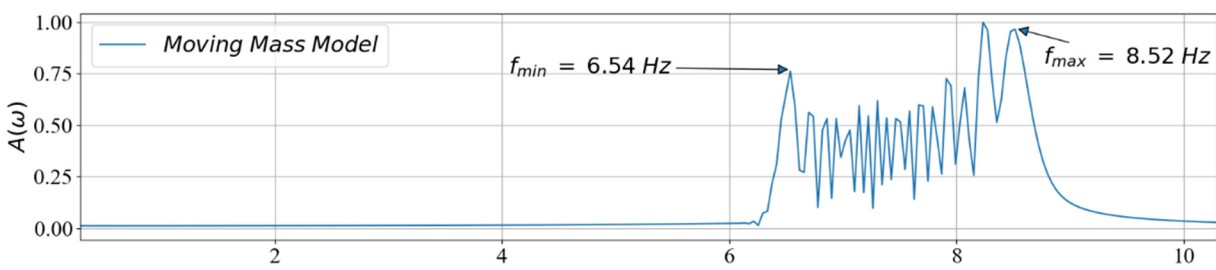

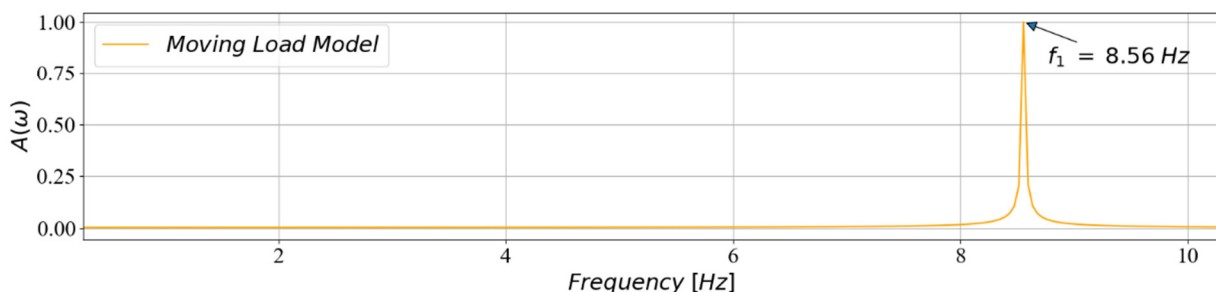

**Figure 7.** FFT of the displacement time history at station $x = 7L/16$ of the bridge deck for the moving mass (**top**) and the moving load (**bottom**) representations of a $m = 38.6$ kg mass speeding at $v = 0.247$ m/s.

## 7. Experimental Validation

Here, we compare the analysis results cast as dimensionless acceleration spectra $A(\omega)$ with those resulting from measurements carried out for the experiment described in Section 1. The basic parameters in the experimental setup were the mass $m = 38.60$ kg traversing the bridge deck velocity $v = 0.247$ m/s. This setup gave a mass ratio of $R = \frac{m}{M} = 0.38$, which is substantial. The left support of the bridge deck, where the mass starts moving, was elastic, and two values were derived from the pad interface to bracket the equivalent spring value as $100 < K_1 < 1000$ (kN/m), while the right support, where the mass exits, was kept rigid, i.e., $K_2 = \infty$. Figure 8 depicts the measurements in both time and frequency domains as registered at the central accelerometer (see Figure 1) from one series of tests that was carried out. Similarly, Figure 9 depicts the FFT of the analytical results derived in the time domain from the numerical solution of Equation (11) for the deck acceleration close to the center span, at $= \frac{7L}{16}$, and normalization by $g$. By comparing analytical with experimental results, we validate the conclusion drawn in the previous section, whereby the simplified model of a moving load is not sufficient to describe the case of a heavy mass traversing a bridge deck unless the moving mass's inertia is accounted for. This results in a structural system whose dynamic properties change as the mass moves across, leading to diffusion of the eigenfrequencies that are activated by the deck vibrations. After passage of the moving mass, the bridge model regains its original structural properties.

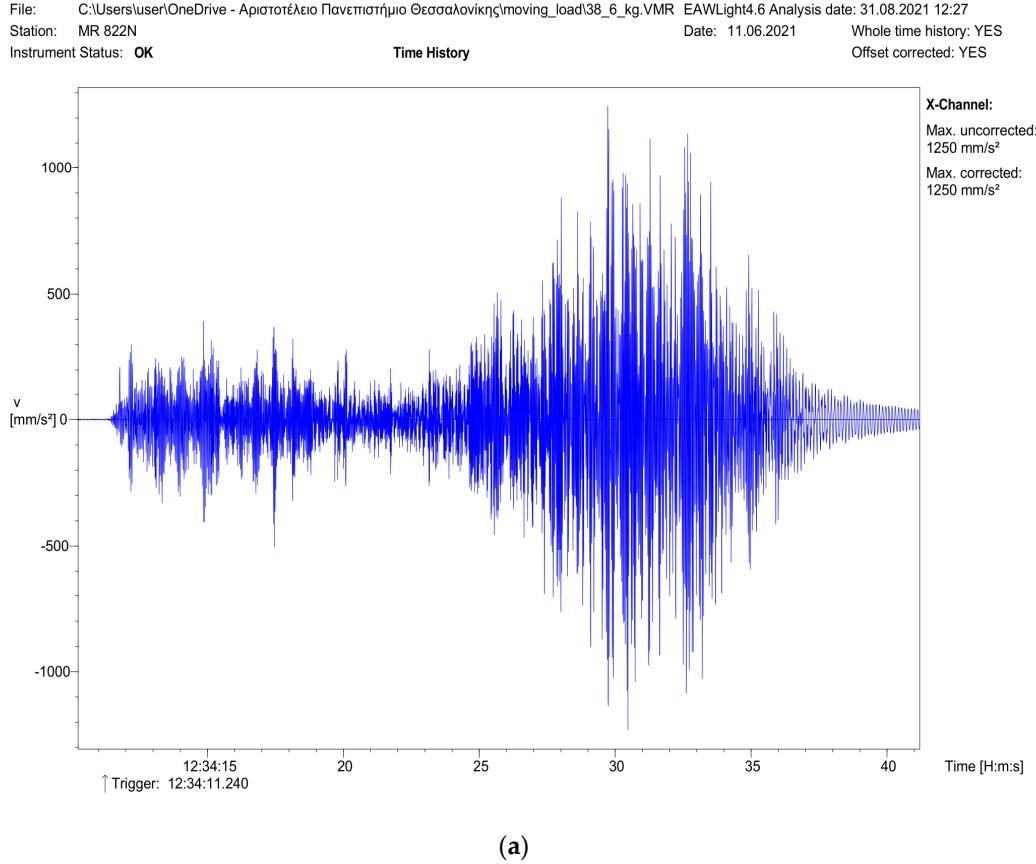

(**a**)

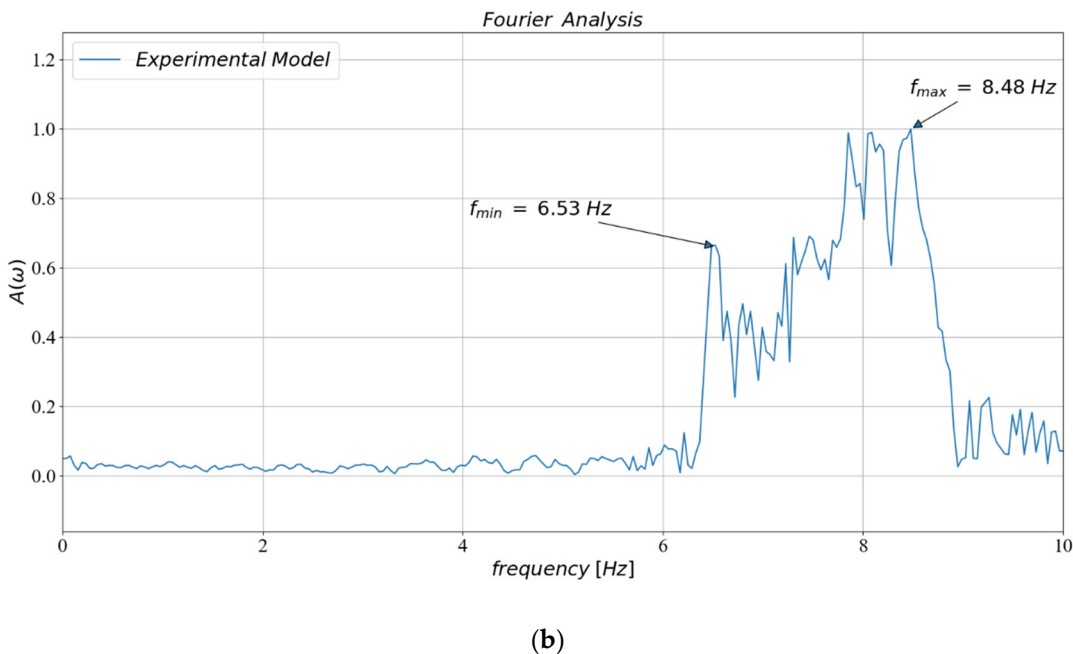

(**b**)

**Figure 8.** Measured (**a**) acceleration time history and (**b**) FFT acceleration spectrum from the experimental setup of a model bridge deck traversed by heavy mass $m = 38.60$ kg moving at a speed of $v = 0.247$ m/s.

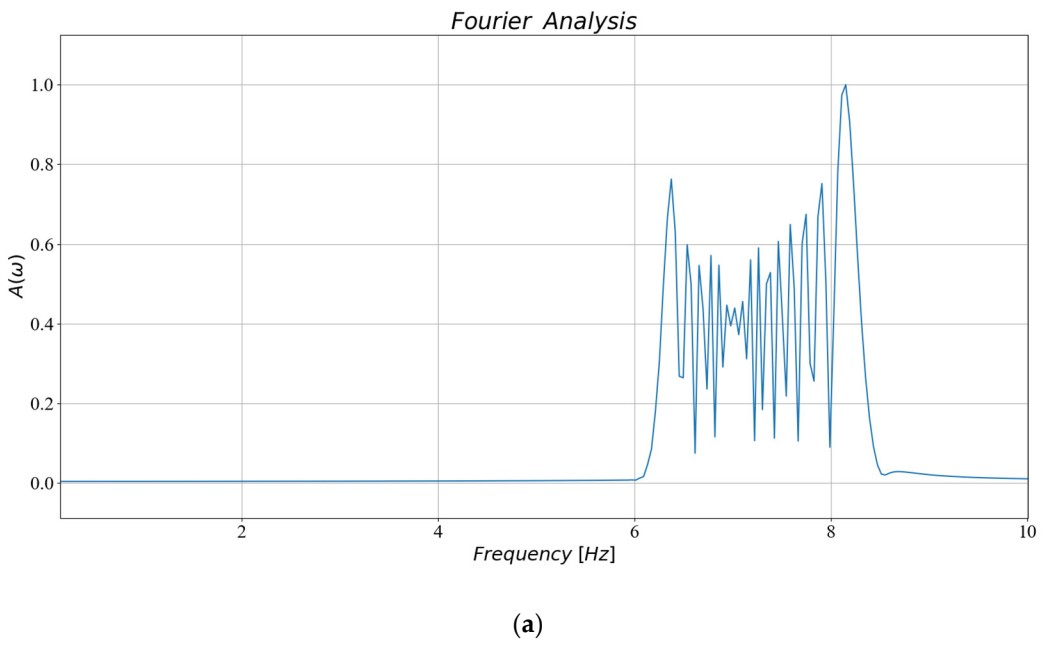

(**a**)

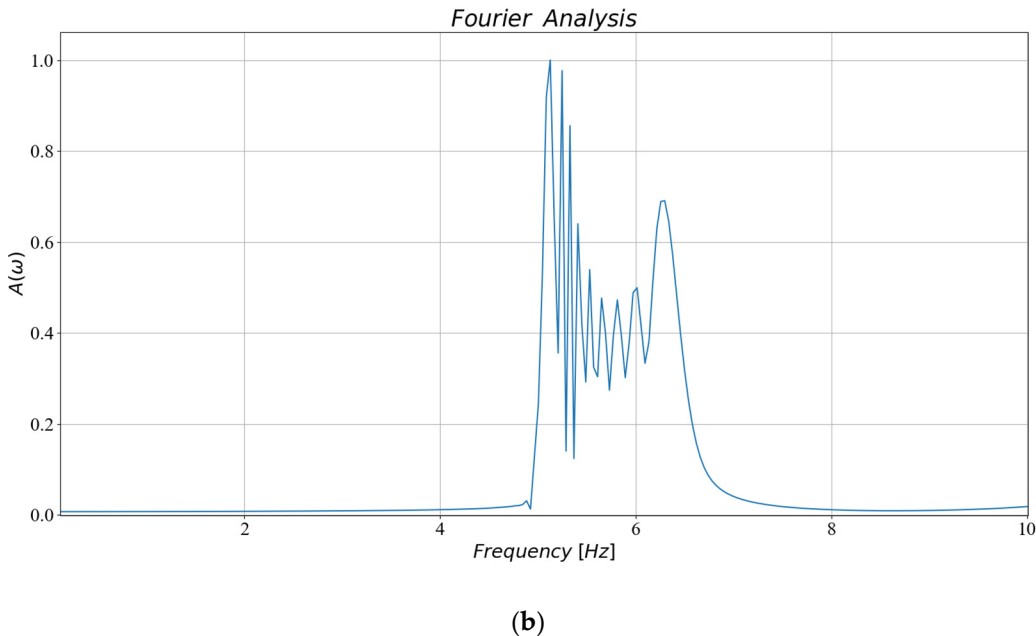

(**b**)

**Figure 9.** Dimensionless acceleration $A(\omega)$ derived from the analytical solution for a heavy mass $m = 38.60$ kg traversing the model bridge with speed $v = 0.247$ m/s and end support stiffness of (**a**) $K_1 = 1000$ kN/m, $K_2 = \infty$. (**b**) $K_1 = 100$ kN/m, $K_2 = \infty$.

*Discussion of the Results*

At first, when the time history plots for the bridge under a rolling mass were transformed to the frequency domain (see Figure 7), we observed diffusion of the bridge eigenfrequencies when the moving mass was large compared to the mass of the bridge deck. Furthermore, when the moving mass was small, i.e., we essentially had a secondary system rolling over a primary system, the dynamic properties of the combined system remained unaltered. This leads to a categorization of bridges as large-scale ones, such as suspension bridges, where traffic is a stream of vehicles whose individual weight is

practically insignificant, versus small-scale ones, such as pontoons, which are crossed by heavy vehicles, with railway bridges falling somewhere in between.

Looking at the results presented in Figure 9 in the frequency domain, we observed two scenarios of stiffness degradation for the left support, namely $K_1 = 1000$ kN/m and $K_1 = 100$ kN/m (with $K_2 = \infty$ in both cases), that showed the influence of the moving mass inertia. The extent of eigenfrequency diffusion was clearly visible in both cases, and a conclusion that can be drawn is the larger the support deterioration, the smaller the width of the resulting diffusion. Observations such as these are, of course, not limited to support degradation, but other factors can be identified that might play a role in bridge deck damage over time that could be of interest in SHM.

Another consequence of the aforementioned eigenfrequency diffusion is the difficulty in establishing the critical velocity of a moving heavy mass. Ordinarily, the relation used is $\omega_1 = (\pi/L)\, v_{cr}$, but now, the first eigenfrequency $\omega_1$ of the bridge is not clearly defined but is instead a swarm of values that has moved to the left in the frequency spectrum, i.e., to lower values. This may lead to an over-prediction of this critical velocity value that may have design consequences for a bridge.

Finally, the potential use for these FFT plots is in an artificial intelligence environment for the development of algorithms that can be used to help detect damage in bridges. Of course, this is a vast area that has been developing over recent decades. However, in the case of our model experiment, the role of the supports turned out to be pivotal in terms of the bridge's eigenproperties. Two simple rules emerge: (a) a shift of the eigenfrequencies to the left in a frequency diagram denotes a more flexible system, possibly indicating support stiffness deterioration; (b) diffusion of the eigenfrequency peaks may indicate residual damage due to the passage of heavy, quickly moving traffic.

## 8. Conclusions

In this work, we developed an analytical solution for a heavy mass rolling with constant speed over the span of a simply supported beam. Following numerical implementation, the results recovered were validated against measurements from a scaled bridge experiment. In reference to the results obtained herein, which we view as forming the background for SHM applied to bridges, we can now distinguish between the following problem parameters: (a) If the moving mass is small, we essentially have a secondary system placed on a primary system, which is the bridge. In this case, the interaction phenomenon is incomplete in the sense that the dynamic properties of the secondary system do not influence those of the primary system. This does not hold true if the moving mass is comparable to that of the bridge it traverses. In that case, the dynamic properties of the primary system are time dependent for the forced vibration period, and once the moving mass leaves the bridge's span, they revert to their original values. Viewed differently, during the forced vibration regime, the eigenvalue problem for the bridge is time dependent. Past that, it becomes the classical eigenvalue problem when the free vibration regime occurs. (b) The other important factor is the speed of the moving mass. Again, we distinguish two regimes, one when the moving mass speed is less than a critical speed that depends on the bridges' material parameters and geometry and another where it is greater. If one looks at the frequency content of a key bridge variable, such as the center span displacement, the peaks corresponding to the eigenfrequencies of the bridge become blurred as the speed of the mass increases and higher modes of vibration are activated. At low speeds and for small masses, these frequency spectra clearly show the peaks corresponding to the eigenfrequencies of the bridge. (c) Finally, at very low speeds, we have a quasi-static problem that yields influence lines, i.e., the deflection curves of the bridge for a given mass position. In closing, all these problem parameters are important in deciding how to implement SHM strategies for bridges, which can be generalized to include railway, highway and overpass bridges.

**Author Contributions:** Conceptualization, G.D.M. and G.I.D.; Project Administration: G.D.M.; Formal Analysis: G.D.M.; Methodology: G.I.D. and G.D.M.; Writing-review and editing: G.D.M.

and G.I.D.; Software: G.I.D.; Validation: G.I.D. and G.D.M. All authors have read and agreed to the published version of the manuscript.

**Funding:** This research was funded by the Deutsche Forschungsgemeinschaft (DFG) grants SM 281/20-1 and SM281/14-1.

**Data Availability Statement:** Data can be made available to interested parties upon request by contacting the authors through the Institute's website https://strength.civil.auth.gr.

**Conflicts of Interest:** The authors declare no conflict of interest.

## Appendix A. The Eigenproblem for Flexible end Supports

When the supports are flexible and represented by the spring constants $K_1$, $K_2$ (kN/m), the eigenfunctions are given as

$$\Phi_n(x) = c_{1n}\left(\sin(k_n x) + \frac{c_{2n}}{c_{1n}}\cos(k_n x) + \frac{c_{3n}}{c_{1n}}\sinh(k_n x) + \frac{c_{4n}}{c_{1n}}\cosh(k_n x)\right)$$
$$n = 1, 2, \ldots$$

where constants $c_{1n}$ are evaluated so that the generalized mass corresponding to each mode is of unit value. Furthermore, $k_n$ is the wave number resulting from setting the $4 \times 4$ system determinant that results from imposing boundary conditions to Equation (2). Specifically, at ends $x = 0, \ L$, we have

$$EI\Phi'''(0) + K_1\Phi(0) = 0, \ \Phi''(0) = 0, \ EI\Phi'''(L) + K_2\Phi(L) = 0, \ \Phi''(L) = 0$$

The determinant thus formed is

$$det = \begin{vmatrix} EIA_1'''(0) + K_1A_1(0) & EIA_2'''(0) + K_1A_2(0) & EIA_3'''(0) + K_1A_3(0) & EIA_4'''(0) + K_1A_4(0) \\ A_1''(0) & A_2''(0) & A_3''(0) & A_4''(0) \\ A_1''(L) & A_2''(L) & A_3''(L) & A_4''(L) \\ EIA_1'''(L) - K_2A_1(L) & EIA_2'''(L) - K_2A_2(L) & EIA_3'''(L) - K_2A_3(L) & EIA_4'''(L) - K_2A_4(L) \end{vmatrix} = 0$$

where

$$A_1(x) = \sin(kx), A_2(x) = \cos(kx), A_3(x) = \sinh(kx), A_4(x) = \cosh(kx),$$

When the above determinant is set equal to zero, the roots of this determinant when set equal to zero are traced using the Newton–Raphson method, followed by back substitution in the expression for the eigenfunctions. The results are plotted in Figure 2 for two basic cases and help determine the placement of the accelerometers along the span in the subsequent experimental procedure.

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
