# Peer review of "Model Bridge Span Traversed by a Heavy Mass: Analysis and Experimental Verification"

_infrastructures, doi:10.3390/infrastructures6090130_

Round 1

Reviewer 1 Report

This is an interesting work.

The manuscript is sound and acceptable for publication once that the followings points are consider:

In section 2. Experimental Set up

- Please mention the value of A and I.

- Can the authors provide more details on the method at which the mass superposition and moving at the experimental setup?

- What are the possible mass and velocity ranges that can be implemented in an experiment?

- Why were the values of K1 and K2 selected as 1000kN/m and 100kN/m, and what is the contribution of these values in this paper?

In section 7. Experimental Validation

- Please include the acceleration data measured related in Figure 8 and 9.

Reviewer 2 Report

The manuscript "Model Bridge Span Traversed by a Heavy Mass: Analysis and Experimental Verification" adheres to the theme of this journal, however many corrections are needed:

a) The abstract is very generic, the innovation/contribution of this study in the literature must be presented, as well as clear quantitative results;
b) In general, the document is very similar to a technical report, which differs from a scientific article. Note that there are only 13 references, and some are not even scientific articles, this should be greatly improved;
c) Discussions are very limited and comparisons with other studies in the literature are lacking, the authors should dedicate themselves to this activity!!!

Reviewer 3 Report

This paper reports the dynamic response of a model bridge under a heavy load with a constant velocity. The quality of this paper is good. Please see the comments below.

  1. the mathematical derivation of this paper takes too much. To make this more concise, the author should focus on the essential parts. So please revise it.
  2. There are many journal papers need be cited in this paper. For example:

Green, M. F., and D. Cebon. "Dynamic interaction between heavy vehicles and highway bridges." Computers & structures 62.2 (1997): 253-264.

Green, M. F., and D. Cebon. "Dynamic response of highway bridges to heavy vehicle loads: theory and experimental validation." Journal of Sound and Vibration 170.1 (1994): 51-78.

Tang, Q., Du, C., Hu, J., Wang, X., & Yu, T. (2018). Surface rust detection using ultrasonic waves in a cylindrical geometry by finite element simulation. Infrastructures, 3(3), 29.

Do, Tin V., Thong M. Pham, and Hong Hao. "Dynamic responses and failure modes of bridge columns under vehicle collision." Engineering Structures 156 (2018): 243-259.

Azim, Md Riasat, and Mustafa Gül. "Data-driven damage identification technique for steel truss railroad bridges utilizing principal component analysis of strain response." Structure and Infrastructure Engineering 17.8 (2021): 1019-1035.

  1. Moderate English change is required.

Round 2

Reviewer 1 Report

I accept in present form

Reviewer 2 Report

ok

Reviewer 3 Report

The author corrected all comments suggested by the reviewer. I recommend this paper be published in the present form.